# MARS: Harmonizing Multimodal Convergence via Adaptive Rank Search

## Abstract

Fine-tuning Multimodal Large Language Models (MLLMs) with parameter-efficient methods like Low-Rank Adaptation (LoRA) is crucial for task adaptation. However, imbalanced training dynamics across modalities often lead to suboptimal accuracy due to negative interference, a challenge typically addressed with inefficient, heuristic methods like manually tuning separate learning rates. To overcome this, we introduce **MARS** (**M**ultimodal **A**daptive **R**ank **S**earch), an approach to discover optimal rank pairs that balance training dynamics while maximizing performance. Our key innovation, a proposed framework of dual scaling laws, enables this search: one law models module-specific convergence time to prune the search space to candidates with aligned dynamics, while the other predicts final task performance to select the optimal pair from the pruned set. By re-purposing LoRA rank as a controller for modality-specific convergence speed, MARS achieves superior performance over baseline methods and offers a robust, automated strategy for optimizing MLLM fine-tuning.

## 1 Introduction

A prominent trend in modern Multimodal Large Language Model (MLLM) research is the shift toward comprehensive fine-tuning of all major components—including the modality encoder (ME), projector, and LLM backbone—to achieve state-of-the-art performance (Zhang et al., 2024b; Zhai et al., 2024; Zanella & Ben Ayed, 2024; Chen et al., 2025). This paradigm shift stems from the growing recognition that simply connecting a modality encoder to a pre-trained LLM is insufficient for unlocking deeply integrated multimodal understanding (Kim et al., 2024). Given the immense scale of these models, this comprehensive adaptation is almost exclusively enabled by parameter-efficient fine-tuning methods like Low-Rank Adaptation (LoRA) (Hu et al., 2022).

While applying a uniform LoRA rank is common practice, this approach overlooks the distinct learning requirements of each modality and fails to address the critical issue of imbalanced training dynamics, where modules converge at different rates. As illustrated in Figure 1, this imbalance can lead to performance bottlenecks and training oscillations. A common alternative, heuristically tuning differential learning rates, is often laborious and relies on costly trial-and-error experimentation (Li et al., 2024; Bai et al., 2025; Zhang et al., 2024b). Given the learning rate only controls the speed of learning via gradient scaling, a more fundamental strategy is to adjust the LoRA rank, which directly controls a module's adaptation capacity and also serves as a regularizer (Biderman et al., 2024). Using differential ranks therefore provides a systematic way to harmonize multimodal fine-tuning. However, identifying an optimal rank pair is challenging due to an inherent *two-fold disparity*: (1) a disparity in learning capacity, stemming from their differing parameter scales, and (2) a disparity in the required learning budget, as each module originates from a distinct pre-trained unimodal model with its own domain gap to the downstream task. Therefore, the core challenge lies in quantifying these disparities to align the convergence dynamics of all modules.

To address this, we introduce **MARS** (**M**ultimodal **A**daptive **R**ank **S**earch), an effective and efficient procedure for identifying optimal rank pairs that ensure aligned training dynamics. The search problem is inherently difficult: the combinatorial space of rank pairs is vast, and each candidate requires full fine-tuning runs to evaluate performance, rendering naive search inefficient and impractical. To make MARS feasible prior to full fine-tuning, we draw inspiration from research on scaling laws, which have proven effective in predicting the capabilities and performance of large deep learning

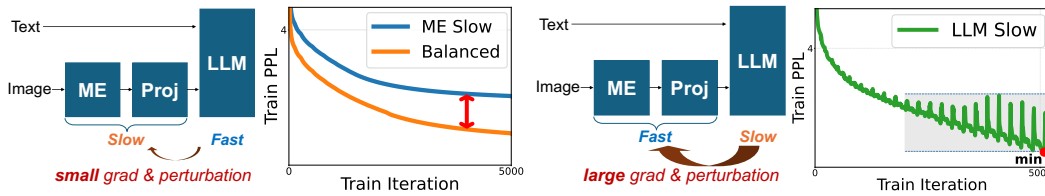

(a) ME Under-adapted    (b) Performance Bottleneck    (c) LLM Under-adapted    (d) Training Oscillation

Figure 1: **Motivation: Imbalanced training dynamics lead to suboptimal performance.** Conceptual cases of module imbalance (**a, c**) and their corresponding empirical results (**b, d**). When the modality encoder (ME) is under-adapted, a performance bottleneck occurs (**b**). When the LLM is under-adapted, it causes training oscillations (**d**).

models (Kaplan et al., 2020; Zhang et al., 2024a; Shukor et al., 2025). These predictive models motivate a systematic data-driven approach over heuristic trial-and-error tuning. We propose dual scaling laws tailored for MLLM fine-tuning: Scaling Law-P (Performance), which predicts final task accuracy and serves as the objective function, and Scaling Law-C (Convergence), which estimates the training iterations required for each module to converge. MARS leverages these laws to prune the search space to candidates with aligned convergence and then selects the optimal rank pair based on predicted performance, thereby substantially reducing search costs while improving results.

Our main contributions are:

- We identify and provide evidence that the imbalanced training dynamics in MLLM fine-tuning, originating from a two-fold disparity, represent a key source of suboptimal performance. To overcome this, we propose MARS, an automated algorithm that systematically mitigates the imbalance by discovering optimal, modality-specific LoRA rank pairs.

- We are the first to propose and validate dual scaling laws for MLLM fine-tuning that model task performance (Scaling Law-P) and module-specific convergence cost (Scaling Law-C), making the rank search feasible.

- Our evaluation demonstrates that models fine-tuned with MARS achieve higher task performance than other baselines, improving ScienceQA accuracy by up to 12.0% and reducing LLaVA Bench perplexity by up to 13.2%. Furthermore, MARS shortens the total time required for search and fine-tuning by over 11.5x compared to baseline search methods.

## 2 MARS METHODOLOGY

**Motivation.** The fine-tuning of MLLMs is often hampered by the imbalanced training dynamics between its constituent modules, leading to suboptimal performance. To clearly isolate and demonstrate this phenomenon, as illustrated in Figure 1, we designed a controlled experimental setting. While in practice these dynamics are caused by an inherent *two-fold disparity*, we sought to show the direct impact of the imbalance itself. To achieve this, we assembled a LLaVA-OneVision-0.5B* model from pre-trained unimodal components that have similar parameter counts and no prior exposure to the multimodal task. In this controlled environment, where both modules must learn the task knowledge, we intentionally induced an imbalance using differential learning rates. Setting a much lower learning rate for the ME ($lr_{me} \ll lr_{llm}$) resulted in a clear performance bottleneck, whereas a much lower learning rate for the LLM ($lr_{llm} \ll lr_{me}$) caused significant training instability.

The standard heuristic to mitigate this is to manually tune differential learning rates; however, this process is laborious, relying on extensive trial-and-error rather than a predictive model. While one could attempt to automate this learning rate search, we argue that a more fundamental approach lies in leveraging a more direct control available during parameter-efficient fine-tuning: *the LoRA rank*. Unlike the learning rate, which simply scales gradient updates, the LoRA rank determines the intrinsic capacity of the adaptation and acts as a powerful regularizer (Biderman et al., 2024). We posit that finding an optimal pair of differential LoRA ranks is therefore a more effective approach to harmonizing multimodal fine-tuning. However, a naive (exhaustive) search for this optimal rank

---

**Algorithm 1:** Comparison of Naive Search and MARS

---

**Input** : Candidate ranks $R_{options}$, target dataset $D_f$
**Output:** Optimal config $(r^*_{me}, r^*_{llm})$
$BestPerformance \leftarrow -\infty$
$CandidatePairs \leftarrow \emptyset$
**for** $r_{llm} \in R_{options}$ **do**

> \# --- Naive Search: Iterate over all rank pair combinations; full fine-tune to obtain accuracy.---
> **for** $r_{me} \in R_{options}$ **do**
> > $Perf \leftarrow$ RUNFULLFINE-TUNING$(r_{me}, r_{llm}, D_f)$
> > **if** $Perf > BestPerformance$ **then**
> > > $BestPerformance \leftarrow Perf$
> > > $(r^*_{me}, r^*_{llm}) \leftarrow (r_{me}, r_{llm})$

> \# +++ MARS: No inner loop; Prunes search space & Selects best ME rank $(r'_{me})$ for $r_{llm}$.+++
> $r'_{me} \leftarrow$ SOLVEBALANCECONDITION$(r_{llm})$        /\* Scaling Law-C \*/
> Add $(r'_{me}, r_{llm})$ to $CandidatePairs$
> $(r^*_{me}, r^*_{llm}) \leftarrow$ SELECTBEST$(CandidatePairs)$        /\* Scaling Law-P \*/

**return** $(r^*_{me}, r^*_{llm})$

---

pair is computationally prohibitive, as each combination would require a full fine-tuning run. In this section, we introduce MARS, which addresses this challenge. We first provide an overview of our methodology, highlighting the dual scaling laws, before elaborating on the detailed search procedure.

## 2.1 MULTIMODAL ADAPTIVE RANK SEARCH

The MARS framework is designed to transform the intractable search for optimal LoRA ranks into an efficient, guided procedure. It functions as a pre-fine-tuning stage prior to final task-specific training. As illustrated in Algorithm 1, a naive search would require a costly inner loop, executing a full fine-tuning run for every possible rank combination. MARS fundamentally improves this process by replacing this expensive inner loop with an efficient, predictive model.

The core of the search is a guided, two-step process that efficiently identifies the optimal rank pair:

1. **Pruning via Convergence Balancing:** First, MARS uses our convergence law (*Scaling Law-C*) to enforce a balance condition ($t_{me} \approx t_{llm}$). This allows it to drastically prune the vast search space by generating a small set of candidate pairs that are predicted to have stable, harmonized training dynamics.

2. **Selection via Performance Prediction:** Subsequently, from this pruned set of stable candidates, MARS uses our performance law (*Scaling Law-P*) to predict the final task accuracy for each pair and selects the one with the best predicted outcome.

Once a fine-tuning job is initiated, the procedure begins with a lightweight calibration phase, where a small set of experiments is conducted to empirically fit the coefficients of the dual scaling laws for the given context. With these coefficients in place, the framework transitions into the search phase, replacing exhaustive evaluations with predictive modeling. This two-phase design enables MARS to efficiently identify a high-performing, convergence-aware rank pair before launching the final full-scale fine-tuning run.

## 2.2 DUAL SCALING LAWS: SCALING LAW-P AND SCALING LAW-C

Dual scaling laws are the predictive foundation that makes the rank search feasible, transforming an otherwise intractable brute-force search into an efficient, guided process. In this section, we elaborate on the details of the dual scaling laws, including how they are formulated and validated in the MLLM fine-tuning setup.

### 2.2.1 EXPERIMENTAL SETUP

To empirically derive our scaling laws, we designed a controlled experimental setup to isolate and measure pure fine-tuning capability. Our experiments are based on the LLaVA-OneVision (OV) architecture (Li et al., 2024), but we initialize our models "from scratch." This means we assemble publicly available modality encoder (ME) (SigLIP; Zhai et al. (2023)) and LLM (Qwen2; Wang et al. (2024)) checkpoints with a pre-trained projector, ensuring the model architecture is consistent while the initial parameters have no prior exposure to the downstream fine-tuning datasets. This controlled initialization allows us to focus solely on the scaling dynamics of the fine-tuning process itself. To study the impact of parameter disparity, we used two variants: LLaVA-OV-0.5B* (minimal ME-LLM parameter gap, with a $\sim$400M ME and $\sim$500M LLM) and LLaVA-OV-7B* (significant gap, with a $\sim$400M ME and $\sim$7B LLM).

Our primary fine-tuning dataset was LLaVA-158K (Liu et al., 2023; Li et al., 2024), from which we created subsets of varying sizes to study scaling trends. To measure the fundamental fine-tuning capability (i.e., transferability) of a given configuration, we use perplexity on a fixed validation set as our primary metric. As perplexity is the exponential of the train loss, it provides a direct and general measure of predictive performance that aligns closely with the training objective. Further details on dataset splits and task composition are provided in Appendix A.

### 2.2.2 SCALING LAW-P FOR PERFORMANCE PREDICTION

**Formulation.** Based on our empirical findings and previous scaling laws on LLM fine-tuning (Zhang et al., 2024a), we formulate Scaling Law-P to model the fine-tuning loss (or perplexity, $\hat{L}$) for an MLLM. This law extends existing frameworks by incorporating separate LoRA ranks for the modality encoder ($r_{\text{me}}$) and the LLM ($r_{\text{llm}}$), alongside the dataset size ($D_f$):

$$\hat{L}(r_{\text{me}}, r_{\text{llm}}, D_f) = A \cdot \frac{1}{(r_{\text{me}})^{\alpha_m} \cdot (r_{\text{llm}})^{\alpha_l} \cdot D_f^{\beta}} + E \tag{1}$$

Here, $\alpha_m$ and $\alpha_l$ are scaling exponents for the ME and LLM ranks, reflecting their impact on loss reduction, while $\beta$ is the exponent for dataset size. $A$ and $E$ are fitted constants representing a scaling coefficient and an irreducible error, respectively. This law serves as the objective function in MARS.

**Key Observation.** We observed critical scaling behaviors differing from unimodal work, motivating us to adapt the existing scaling law into Scaling Law-P for the MLLM fine-tuning setting.

- **Performance is sensitive to the interplay between ME and LLM ranks.** Departing from unimodal studies where LoRA rank often has a modest impact, we find that in MLLM fine-tuning, performance is highly dependent on the combination of ranks chosen for the modality encoder (ME) and the LLM. An improper balance between these ranks can negate the benefits of training, even on large datasets (Figure 2 (a)). This suggests that both ranks should be considered jointly, as their interplay is a key factor for effective fine-tuning.

- **Optimal rank involves a trade-off with dataset size.** The relationship between LoRA rank and performance is not monotonic. While small datasets tend to benefit from higher ME ranks to achieve better feature extraction (Figure 5 in Appendix), large datasets achieve better results with more moderate ranks to avoid overfitting. Therefore, the optimal rank must be selected by balancing the need for expressive feature learning against the risk of overfitting, a consideration that is tied to the dataset size.

These observations collectively underscore the need for an adaptive, modality-conscious rank selection strategy. Additional analysis is presented in Appendix C.

### 2.2.3 SCALING LAW-C FOR CONVERGENCE SPEED PREDICTION

**Formulation.** Modeling the relationship between LoRA rank, dataset size, and the number of iterations required for convergence is a relatively underexplored area. To address this, we propose Scaling Law-C to model the convergence cost for each modality-specific module within an MLLM.

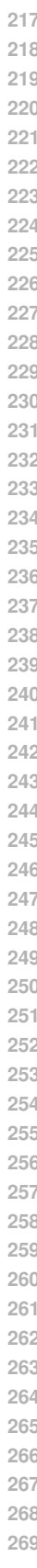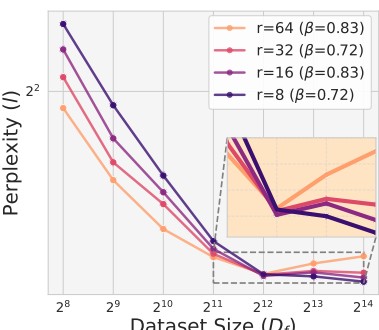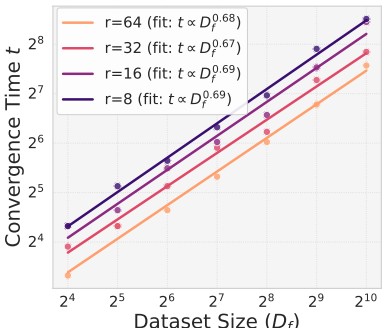

(a) Scaling Law-P: Performance (perplexity) as a function of dataset size for different LLM ranks ($r_{me} = 4$). The yellow line ($r_{llm} = 64$) shows that larger discrepancies between ME and LLM lead to reduced fine-tuning capability.

(b) Scaling Law-C: Convergence time as a function of dataset size for different LLM ranks ($r_{me} = 16$). The near-parallelism of the fitted lines supports the separability of the rank and dataset size terms in Equation 2.

Figure 2: The proposed dual scaling laws from LLaVA-OV-0.5B* (ME and LLM have a similar parameter count without task-specific knowledge). This pattern holds consistently across different fixed-rank settings (e.g., $r_{me} = 8$ and $r_{me} = 32$)

For a given module $i$, where $i$ can represent a modality encoder (ME) or the LLM, we define the number of training steps to convergence, $t_i$, with the following general form:

$$t_i(r_i, D_f) = k_i \cdot (r_i)^{\gamma_i} \cdot D_f^{\delta_i} + E_i \tag{2}$$

In this formulation, $t_i$ is the predicted training steps to convergence for module $i$ (e.g., $i \in \{me, llm\}$). The LoRA rank for this module is $r_i$, and $D_f$ is the fine-tuning dataset size. The exponent $\gamma_i$ links the LoRA rank to convergence speed; based on our observations that higher ranks reduce the number of steps, we fit this as a negative value. The exponent $\delta_i$ quantifies the impact of dataset size on convergence steps. Finally, $k_i$ and $E_i$ are constants fitted from experimental data for each module. This general formulation allows us to predict how the convergence cost of any modality scales with its allocated LoRA rank and the amount of training data, which is crucial for the pruning stage of MARS where we aim to balance $t_{me}$ and $t_{llm}$.

**Key Observation.** Our empirical investigations provide strong support for the proposed formulation of Scaling Law-C, revealing consistent trends in how convergence time (measured by the number of training iterations to reach minimum test perplexity) scales with dataset size and LoRA rank.

- **Increasing dataset size *increases* convergence time.** Larger fine-tuning datasets ($D_f$) demand substantially more iterations for the model to reach convergence. This is intuitive, as larger datasets contain more information, and the model requires more training steps to adequately process and learn from these additional samples to reach an optimal state. This near-linear trend, illustrated in Figure 2 (b), aligns with the $D_f^{\delta_i}$ term in Equation 2.

- **Increasing rank size *decreases* convergence time.** Modules with higher LoRA ranks ($r_i$) converge in fewer iterations, as larger ranks provide the adapters with greater capacity and flexibility to adapt efficiently during fine-tuning. As a result, these modules reach convergence faster than those with lower, more constrained ranks. As shown in Figure 2 (b), this inverse relationship aligns with the $r_i^{\gamma_i}$ term in Equation 2, where a negative $\gamma_i$ exponent captures the observed acceleration.

These two distinct scaling behaviors form the empirical basis for our convergence speed predictions, which are integral to the MARS algorithm's ability to identify convergence-balanced rank pairs.

## 2.3 THE GUIDED SEARCH SPACE OPTIMIZATION

Once the dual scaling laws are calibrated, MARS performs its automated search for the optimal LoRA rank pair $(r_{\text{me}}^*, r_{\text{llm}}^*)$ for a given target dataset size $D_{\text{target}}$. The search first prunes the vast space of possible rank combinations using our fitted Scaling Law-C. The guiding principle is to identify rank pairs where the ME and LLM modules are predicted to have aligned convergence times ($t_{\text{me}} \approx t_{\text{llm}}$). By solving this balance condition, we can express the ideal ME rank as a direct function of the LLM rank:

$$r_{\text{me}} \approx \left( \frac{k_l \cdot (r_{\text{llm}})^{\gamma_l} \cdot D_f^{\delta_l} + E_{\text{llm}} - E_{\text{me}}}{k_v \cdot D_f^{\delta_v}} \right)^{\frac{1}{\gamma_v}} \tag{3}$$

To generate a set of *convergence-aligned* candidates, we iterate through a list of possible LLM ranks, use Equation 3 to calculate the corresponding ideal ME rank for each, and map it to the closest available discrete rank. This procedure efficiently creates a pruned set of candidate pairs, `CandidatePairs` in Algorithm 1.

The final step is to select the pair with the highest performance potential. We use the fitted Scaling Law-P to predict the final fine-tuning loss, $\hat{L}$, for each candidate pair, where $\Theta_{\text{acc}}$ denotes the fitted coefficients. The optimal rank pair is the one that minimizes this predicted loss:

$$(r_{\text{me}}^*, r_{\text{llm}}^*) = \underset{(r_{\text{me}}, r_{\text{llm}}) \in \texttt{CandidatePairs}}{\text{argmin}} \hat{L}(r_{\text{me}}, r_{\text{llm}}, D_f; \Theta_{\text{acc}}) \tag{4}$$

The value of our dual scaling law framework lies in its synergistic two-phase approach. A search guided only by a performance predictor can yield *practically unstable* configurations due to the gradient interference caused by imbalanced convergence, while optimizing only for balance does not guarantee the highest accuracy. MARS resolves this trade-off by first using Scaling Law-C to prune the search space to a stable subset of candidates, and then employing Scaling Law-P to select the highest-performing solution from that set.

## 2.4 CALIBRATION OF DUAL SCALING LAWS

When fine-tuning begins, the MARS procedure starts with a calibration phase that empirically estimates the coefficients of our dual scaling laws. To minimize computational cost, we employ an efficient data collection strategy: for each representative LoRA rank pair (e.g., $r_{\text{me}}, r_{\text{llm}} \in \{8, 16, 32, 64\}$), we initiate a single fine-tuning run and record performance and convergence metrics at multiple intermediate checkpoints corresponding to smaller effective dataset sizes (e.g., at $2^{10}$ and $2^{11}$ fine-tuning steps). This approach yields a rich dataset, $\mathcal{D}$, from a minimal number of training runs. Using this dataset, we determine the coefficients for both Scaling Law-P and Scaling Law-C by framing it as a numerical optimization problem. Following established practices (Hoffmann et al., 2022a; Zhang et al., 2024a), we employ the L-BFGS-B algorithm to find the parameter set that minimizes the Huber loss between our model's predictions and the observed data. Crucially, this calibration stage utilizes the same fine-tuning settings (e.g., optimizer, batch size) as the final fine-tuning, ensuring the identified rank pair's effectiveness is transferable. This process culminates in reliable, predictive models for performance and convergence dynamics, which are essential for the subsequent guided search.

## 3 EVALUATION

### 3.1 EVALUATION SETUP

**Evaluation Benchmarks and Datasets.** To evaluate the effectiveness of MARS, we utilize two distinct benchmarks designed to measure different aspects of fine-tuning performance. For general fine-tuning capability and transferability, we use LLaVA Bench(Liu et al., 2023), which consists of 15k validation samples from the LLaVA-Instruct dataset and is composed of complex reasoning tasks. For this benchmark, we use validation perplexity as the primary metric. Perplexity, computed directly from cross-entropy loss, serves as a proxy for the model's training objective (minimizing loss) and provides a direct measure of its fine-tuning capability by indicating how well it can specialize on a downstream dataset. To assess performance on a specialized task with a significant domain

Table 1: Comparison with fixed-rank tuning across different learning rates. We determined the best ME learning rate (★) for each LLM learning rate ($\mathbf{lr_{llm}}$) by selecting the value that yielded either the lowest perplexity or the highest accuracy. For the LLaVA Bench, lower perplexity is better (↓), while for ScienceQA, higher accuracy is better (↑). Best in bold, second-best underlined.

| Model | Benchmark | LoRA ($lr_{me}$, $lr_{llm}$) | | | MARS |
|---|---|---|---|---|---|
| | | (★, 1e-5) | (★, 1e-6) | (★, 1e-7) | |
| LLaVA-OV-0.5B (Li et al., 2024) | LLaVA Bench (↓) | 2.7336 | 2.771 | 2.8472 | **2.7188** |
| | ScienceQA (↑) | 71.06 | 61.88 | 59.28 | **72.85** |
| LLaVA-OV-7B (Li et al., 2024) | LLaVA Bench (↓) | 2.2317 | 2.295 | 2.4346 | **2.1875** |
| | ScienceQA (↑) | 72.26 | 69.86 | 67.27 | **74.25** |
| Qwen2.5-VL-3B (Bai et al., 2025) | LLaVA Bench (↓) | 3.6156 | 3.7415 | 4.1557 | **3.5925** |
| | ScienceQA (↑) | 78.04 | 76.45 | 76.25 | **79.24** |
| Qwen2.5-VL-7B (Bai et al., 2025) | LLaVA Bench (↓) | 3.5032 | 3.5908 | 3.8716 | **3.3879** |
| | ScienceQA (↑) | **79.84** | 76.25 | 74.25 | 79.64 |

gap, we use the ScienceQA benchmark (Lu et al., 2022). This benchmark contains multimodal multiple-choice questions across diverse scientific subjects, including natural science, language science, and social science. For our experiments, we use a 16k training split and evaluate on 500 test samples. Following the standard evaluation protocol, we report performance using accuracy.

**Baselines.** To demonstrate the effectiveness of MARS as an adaptive rank search algorithm for convergence coordination, we compare it against several key baselines. The first is a differential learning rate approach, representing the standard heuristic of manually tuning separate learning rates for the ME and LLM. Second, since we argue that using differential ranks between modalities is important, we compare against a set of fixed differential rank pairs (i.e., uniform but different across modalities). Finally, we benchmark against AdaLoRA (Zhang et al., 2023), a most representative adaptive rank allocation method designed for unimodal models, for which we use the officially released source code (target rank=32; see Appendix B for more details). By comparing against these methods, we can show how MARS effectively finds the best rank combination, leading to superior fine-tuning capability.

**Implementation Details.** Our experiments are implemented using the Cornstarch MLLM training framework (Jang et al., 2025). All models were fine-tuned using the Adam optimizer with a cosine learning rate scheduler, and all experiments were conducted on a single NVIDIA GH200 GPU. Further details regarding specific hyperparameters, LoRA configurations, and the MARS calibration setup are provided in Appendix G.

## 3.2 MAIN RESULTS

We evaluate MARS across multiple MLLM architectures and sizes, comparing it against several strong baselines on the LLaVA Bench and ScienceQA benchmarks. Our results, presented in Table 1 and Table 2, demonstrate that MARS leads to largerly higher performance after fine-tuning.

**Comparison with Differential Learning Rates.** Table 1 compares MARS against the common heuristic of using fixed LoRA ranks while tuning differential learning rates. On the LLaVA Bench, MARS consistently achieves a lower (better) perplexity across all tested models. For instance, on the LLaVA-OV-7B model, MARS achieves a perplexity of 2.1875, outperforming the best-performing differential LR setting (2.295). This trend holds on the more specialized ScienceQA benchmark, where MARS improves the accuracy of the LLaVA-OV-7B model from 72.26% to 74.25%. These results demonstrate that our rank-centric search is a more effective strategy for harmonizing module dynamics than just tuning learning rates.

**Comparison with Other Baselines.** In Table 2, we benchmark MARS against a wider range of fine-tuning strategies. MARS consistently surpasses Full-rank Tuning, highlighting the regularization benefits of our adaptive PEFT approach. More importantly, it outperforms various fixed differential LoRA rank pairs, proving the value of our guided search over simply selecting arbitrary rank combinations. The most critical comparison is with AdaLoRA, a representative adaptive rank method designed for unimodal models. MARS significantly and consistently outperforms AdaLoRA

Table 2: Comparison with adaptive rank search baselines. We determined the best ME rank size (★) for each LLM rank size ($r_{llm}$) by selecting the value that yielded either the lowest perplexity or the highest accuracy. For the LLaVA Bench, lower perplexity is better (↓), while for ScienceQA, higher accuracy is better (↑). Best in bold, second-best underlined.

| Model | AdaLoRA | Full-rank Tuning | LoRA ($r_{me}$, $r_{llm}$) | | | MARS |
|---|---|---|---|---|---|---|
| | | | (★, 8) | (★, 16) | (★, 32) | |
| *LLaVA Bench (perplexity ↓)* | | | | | | |
| LLaVA-OV-0.5B (Li et al., 2024) | 2.8973 | 2.7209 | 2.7582 | 2.7336 | 2.7331 | **2.7188** |
| LLaVA-OV-7B (Li et al., 2024) | 2.5189 | 2.2693 | 2.3181 | 2.2317 | 2.4420 | **2.1875** |
| Qwen2.5-VL-3B (Bai et al., 2025) | 3.6679 | 3.6528 | 3.5912 | 3.6156 | 3.7811 | **3.5825** |
| Qwen2.5-VL-7B(Bai et al., 2025) | 3.6394 | 3.5917 | 3.5011 | 3.5032 | 3.6098 | **3.3616** |
| *ScienceQA bench (accuracy (%) ↑)* | | | | | | |
| LLaVA-OV-0.5B (Li et al., 2024) | 62.28 | 69.66 | 70.46 | 71.06 | 69.86 | **72.85** |
| LLaVA-OV-7B (Li et al., 2024) | 66.27 | 70.46 | 70.26 | 72.26 | 73.65 | **74.25** |
| Qwen2.5-VL-3B (Bai et al., 2025) | 70.06 | 75.25 | 78.24 | 78.04 | 78.04 | **79.24** |
| Qwen2.5-VL-7B (Bai et al., 2025) | 73.85 | 77.45 | **79.84** | 79.84 | 78.24 | 79.64 |

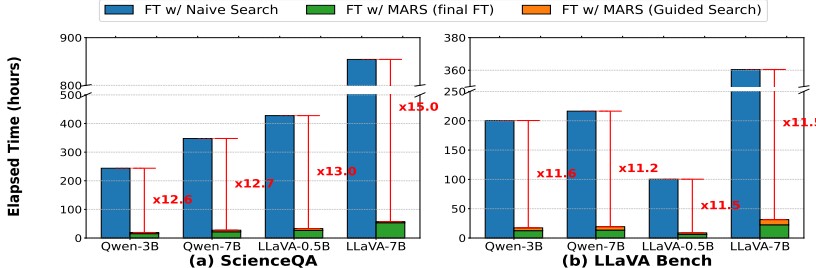

Figure 3: **[Figure Updated]** Time comparison between Naive Search and MARS. Naive Search explores the grid $\{4, 8, 16, 32\} \times \{4, 8, 16, 32\}$ to find the optimal rank pair ($r_{me}, r_{llm}$). MARS accounts for both the search time and the full fine-tuning time of the selected rank pair.

across all models and benchmarks. This confirms that methods designed for single-modality, layer-wise saliency are insufficient for the unique challenge of harmonizing the training dynamics between different modalities in MLLMs.

## 3.3 ANALYSIS

**Computational Efficiency.** Beyond its advantages in fine-tuning effectiveness, MARS also achieves a substantial improvement in computational efficiency. As shown in Figure 3, a naive exhaustive search for the optimal rank pair requires extensive compute, often exceeding 100 GPU hours. In contrast, MARS consists of a lightweight calibration phase followed by a single fine-tuning run, reducing the total elapsed time by more than 11.5x on average across different models and tasks, with most of the time spent on the actual fine-tuning (which is unavoidable). This substantial reduction in search cost demonstrates that MARS is not only more effective, but also more practical and scalable for MLLM fine-tuning.

**Ablation Study: Evaluation on From-Scratch MLLMs.** To assess pure fine-tuning capability, we evaluate task performance on "from-scratch" models whose modules have not been exposed to multimodal training data. Since the original from-scratch models are not publicly available (Li et al., 2024; Wang et al., 2024), we instead assembled unimodal models and connected them with an MLP projector to replicate the

| Benchmark | Model | Best Base. | MARS |
|---|---|---|---|
| LLaVA (ppl ↓) | OV-0.5B* | 4.99 | **4.95** |
| | OV-7B* | 3.62 | **3.56** |
| ScienceQA (% acc. ↑) | OV-0.5B* | 42.49 | **45.01** |
| | OV-7B* | 64.15 | **65.07** |

Table 3: Results on from-scratch cases.

reference architectures (Sec.2.2.1). As shown in Table 3, MARS consistently demonstrates stronger fine-tuning capability than the best baseline configurations (determined by the experiments shown in Tables. 1 and 2). This highlights its effectiveness in enabling from-scratch models to acquire downstream knowledge. See Appendix H for detailed results.

**Ablation Study: Impact of Data Sampling Range.** We conducted an ablation study to determine the optimal range of data required for the lightweight calibration phase of MARS. We ran the MARS search procedure multiple times, each time using a different maximum data sampling range for calibration, and then evaluated the final test perplexity achieved by the resulting optimal rank pair. As shown in Figure 4, we observe that increasing the data sampling range up to $2^{11}$ leads to a clear improvement in the final perplexity, as it allows for a more accurate fitting of the scaling laws. However, performance gains diminish significantly beyond that point, validating our choice of using a data sampling range up to $2^{11}$ for MARS calibration, as it strikes an effective balance between accuracy and computational efficiency.

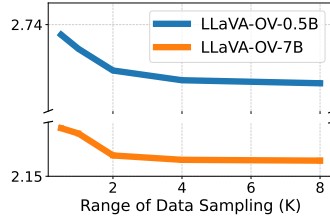

Figure 4: Impact of data sampling range (y-axis: perplexity).

## 4 RELATED WORK

**Fine-Tuning Strategies for MLLMs.** A prominent trend in recent MLLM research is the move towards comprehensive fine-tuning of all major components—including the modality encoder, projector, and LLM backbone—to achieve state-of-the-art performance (Zhang et al., 2024b; Zhai et al., 2024; Zanella & Ben Ayed, 2024; Chen et al., 2025). This paradigm shift away from methods that keep large parts of the model frozen stems from the growing recognition that simply connecting a modality encoder to a static, pre-trained LLM has limitations. Deeper integration, where both modalities can adapt during fine-tuning, is required to unlock more advanced reasoning capabilities (Kim et al., 2024). Given the immense scale of these models, this comprehensive adaptation is almost exclusively enabled by parameter-efficient fine-tuning methods, particularly Low-Rank Adaptation (LoRA) (Hu et al., 2022). In general, prior work adopts a uniform rank across all modules with differential learning rates, and selects the checkpoint that yields the highest accuracy (Liu et al., 2023; Li et al., 2024; Wang et al., 2024; Bai et al., 2025).

**Scaling Laws.** Research on scaling laws, initiated by work on LLM pre-training (Kaplan et al., 2020; Hoffmann et al., 2022b), has established that model performance scales predictably with factors like model size and data. More recent work has extended this analysis to LLM fine-tuning (Zhang et al., 2024a) and the pre-training of native multimodal models (Shukor et al., 2025; He et al., 2025), primarily focusing on predicting final task performance. However, the study of scaling laws in the fine-tuning of pre-trained MLLMs remains largely unexplored.

## 5 CONCLUSION

We introduce MARS to resolve imbalanced convergence speeds in MLLM fine-tuning. MARS employs dual scaling laws to predict convergence and prune the LoRA rank search space, identifying optimal rank pairs with balanced dynamics before actual fine-tuning. With far less search time than a naive approach, MARS demonstrates better fine-tuning capability than common heuristic-based strategies, such as differential learning rates and rank selection, highlighting the importance of convergence coordination for effective multimodal adaptation.

**Limitation & Future Work.** We recognize several limitations and opportunities for future exploration, consistent with the empirical nature of prior scaling law research (Kaplan et al., 2020; Zhang et al., 2024a). Our proposed dual scaling laws are derived primarily from empirical results on open-ended (LLaVA Bench) and closed-generation (ScienceQA) tasks. Additional experiments are needed to determine whether these laws can generalize across all possible MLLM-related scenarios, but such studies exceed our available computational resources. Moving forward, we plan to investigate the theoretical groundings of multimodal scaling behaviors and extend our study to physical multimodal models involving more than two modalities (Kim et al., 2024). In particular, we aim to address the more intricately entangled imbalances in training dynamics that stem from varying domain gaps between pre-training and downstream tasks, as well as from differing learning capabilities across a broader range of modality types and combinations.

## REPRODUCIBILITY STATEMENT

We include implementation details in Section 2.2.1, Appendix A, and Appendix G. The codebase will be open-sourced upon acceptance.

## ETHICS STATEMENT

Our work, multimodal rank search guided by dual scaling laws, aligns with the ICLR Code of Ethics. As MLLMs become more integrated into real-world applications, their adoption through fine-tuning is becoming increasingly popular across a wider range of downstream domains. The optimal rank pair identified by MARS reduces fine-tuning costs by eliminating the repetitive fine-tuning process caused by laborious trial-and-error hyperparameter searches. Ultimately, MARS will accelerate the development cycle and lower the carbon footprint for large-scale MLLM fine-tuning.

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

APPENDIX

# A  EXPERIMENTAL DETAILS FOR SCALING LAW ANALYSIS

## A.1  MODEL SETUP

Our target model architecture is LLaVA-OneVision(LLaVA-OV; Li et al. (2024)). For these scaling law experiments, all models were initialized "from scratch," a methodology we denote with an asterisk (*). This process involves downloading publicly available, pre-trained modality encoder (ME) and Large Language Model (LLM) checkpoints and connecting them with a MLP projector, without any subsequent instruction tuning. Specifically, we used SigLIP (Zhai et al., 2023) for the ME and Qwen2 (Team, 2024) for the LLM and attached a 2-layer MLP projector to replicate the original structure.

This approach is intentionally designed to ensure that the model's initial state is *domain-knowledge-free* with respect to multi-modal instruction data. By doing so, we can isolate and measure the effects of pure fine-tuning capability as a function of data and model scale. We investigated two architectural variants to analyze the impact of the parameter gap between the ME and LLM components.

1. **LLaVA-OV-0.5B\*:** Features a *minimal* parameter gap between its ME and LLM modules (ME ~400M, LLM ~500M).

2. **LLaVA-OV-7B\*:** Features a *significant* parameter gap (ME ~400M, LLM ~7B).

## A.2  DATASET SETUP

Our primary dataset is the LLaVA dataset (Liu et al., 2023; Li et al., 2024), which, to our knowledge, is one of the largest of its kind available for research. Each sample is centered on images of nature or everyday life. The dataset comprises a mixture of open-ended task types, including basic image captioning, detailed image captioning, and multi-turn conversational question answering.

To study scaling trends, we constructed training subsets from this dataset by varying their size according to a power-of-two progression (i.e., $2^n$ samples, for $n$ from 3 to 15), while maintaining a consistent distribution of the above task types across all subsets. For validation, we used a fixed subset of 500 samples (non-overlapping with training data) to enable frequent, low-cost perplexity checks during intermediate training cycles.

# B  ADDITIONAL RELATED WORK

**Adaptive LoRA for Unimodal Models.**  To move beyond static configurations, adaptive LoRA methods like AdaLoRA (Zhang et al., 2023) and SalientLoRA (Ke et al., 2024) were developed. However, these methods are designed for unimodal models and face fundamental limitations in multimodal setups. In the unimodal context, it is often observed that allocating a higher rank to more salient layers correlates with improved transferability and performance (Biderman et al., 2024), paralleling model compression techniques that allocate more resources to higher-importance components (Dong et al., 2019; Lee et al., 2021). In contrast, we find that for MLLMs, simply assigning higher ranks does not invariably lead to better performance. This is due to the complex and crucial inter-modality dependencies, where the optimal configuration for one modality is influenced by the state of the other.

# C  OPTIMAL ME RANK SIZE IN THE SMALL-DATASET REGIME

The optimal ME rank is strongly dependent on the dataset size, revealing a critical trade-off between rapid adaptation and overfitting. As illustrated in the heatmaps (Figure 5), small-dataset regimes clearly prefer higher ME ranks. This is because the ME initially faces a large domain gap, and

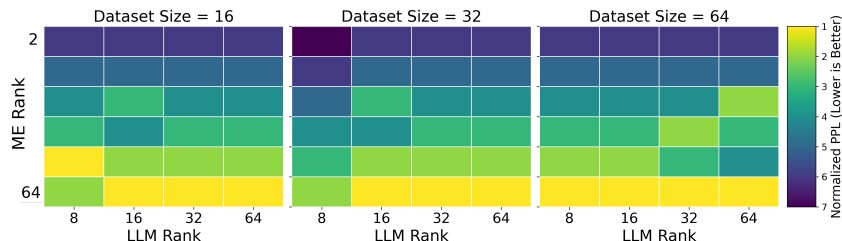

Figure 5: Optimal ME rank size in the small-dataset regime. Lower values (lighter colors) indicate better performance (i.e., lower perplexity).

a higher rank provides the necessary adaptation capacity to quickly align with the LLM. From a capacity perspective, this aggressive approach allows the ME to absorb all available knowledge from limited samples, preventing it from becoming an early learning bottleneck.

Conversely, in mid-to-large dataset regimes, a different pattern emerges: the preference shifts to smaller or more moderate ME ranks. The primary goal becomes reducing overfitting risk and ensuring the efficient utilization of model capacity. With a large and diverse set of examples, extreme adaptation capacity is no longer required. Instead of attempting to absorb all available knowledge, a moderate rank is sufficient to extract essential visual information without over-specializing on training examples. This provides the LLM with stable, moderately adapted inputs, which promotes balanced convergence, prevents overfitting, and allows the model to fully leverage the LLM's capacity for optimal modality harmonization.

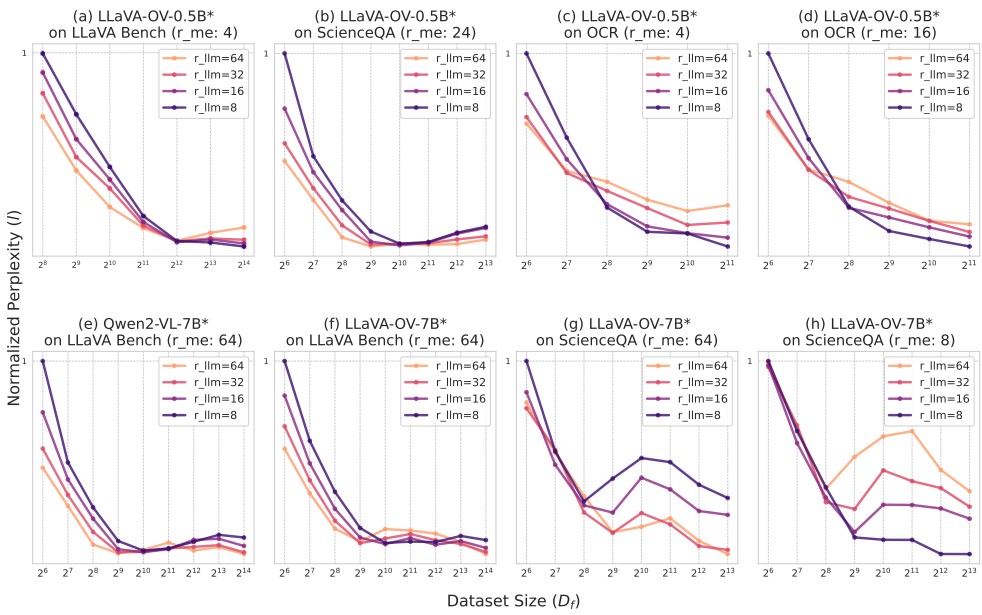

Figure 6: Validating Scaling Law-P Across Different Fine-Tuning Setups

## D  FURTHER VALIDATION ON SCALING LAW-P

Scaling Law-P is grounded in the scaling laws proposed by prior work (Zhang et al., 2024a), which were originally established in the context of LLM fine-tuning. However, our work provides a counter-example: we observed that the original scaling laws hold under ideal conditions; specifically, when there is no imbalance in training dynamics across heterogeneous components. When such an imbalance exists, we consistently observe abrupt increases in perplexity in the large-data regime.

Figure 6 summarizes these findings across multiple architectures (Li et al., 2024; Wang et al., 2024), different task types (LLaVA Bench: open-ended general tasks; ScienceQA and OCR: relatively close-ended domain-specific tasks), and varying learning capacities between the ME and LLM components (LLaVA-OV-0.5B: similar capacities; LLaVA-OV-7B: different capacities).

## E    FURTHER VALIDATION ON SCALING LAW-C

We provide extended validation of Scaling Law-C across different model architectures and parameter scales. Figure 7 shows convergence time as a function of dataset size on a log–log scale for the LLaVA-OV-0.5B* model (fine-tuned on the LLaVA dataset). To isolate the scaling behavior of each module (e.g., ME), we fix the rank of the other module (e.g., LLM rank fixed at 16).

To demonstrate the generality of these findings, we conduct the same validation on the Qwen2.5-VL family (3B and 7B) using the ScienceQA dataset, as shown in Figure 8. Consistent with the LLaVA results, the Qwen plots exhibit a near-linear relationship on the log–log scale. Notably, the fitted lines for different ranks remain nearly parallel across all architectures, providing strong evidence for the separability of the rank and dataset-size terms in our proposed scaling law (Equation 2).

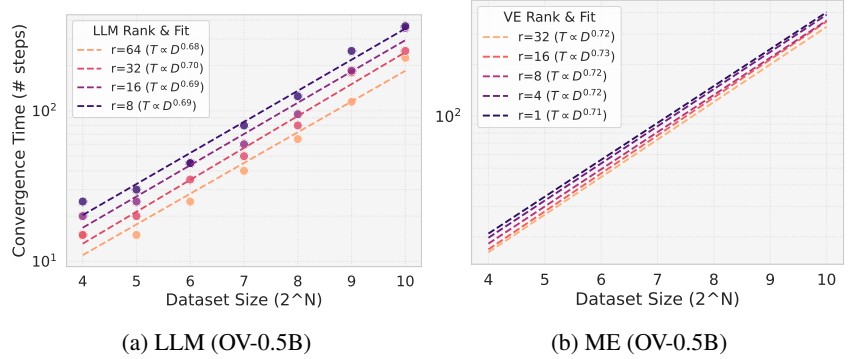

(a) LLM (OV-0.5B)                    (b) ME (OV-0.5B)

Figure 7: Validating Scaling Law-C Across Model Components

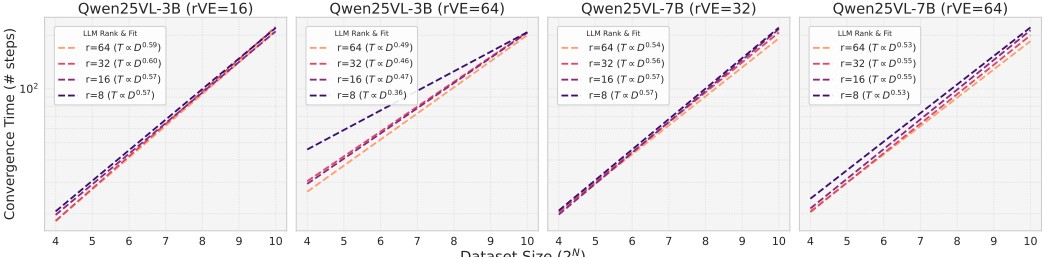

Figure 8: Scaling Law-C Validation on Qwen2.5-VL Models for the ScienceQA Dataset

## F    DERIVATION OF EQUATION 3

This section provides a step-by-step derivation of Equation 3, which estimates the ideal ME-to-LLM parameter ratio ($r_{\mathrm{me}}$). The "ideal" ratio is defined as the value that balances the expected training time of the modality encoder ($t_{\mathrm{me}}$) and the LLM ($t_{\mathrm{llm}}$) for a given target fine-tuning dataset size.

The derivation starts with our two empirically fitted scaling law models for training time:

$$t_{\mathrm{me}} = k_v \cdot (r_{\mathrm{me}})^{\gamma_v} \cdot D_f^{\delta_v} + E_{\mathrm{me}} \tag{5}$$

$$t_{\mathrm{llm}} = k_l \cdot (r_{\mathrm{llm}})^{\gamma_l} \cdot D_f^{\delta_l} + E_{\mathrm{llm}} \tag{6}$$

where $\Theta_{t\_ve} = \{k_v, \gamma_v, \delta_v, E_{\mathrm{me}}\}$ and $\Theta_{t\_llm} = \{k_l, \gamma_l, \delta_l, E_{\mathrm{llm}}\}$ are the fitted coefficients.

To find the balance point, we set these two time estimates to be approximately equal ($t_{\mathrm{me}} \approx t_{\mathrm{llm}}$) for a target dataset size, denoted as $D_{\mathrm{target}}$. This gives the following relationship:

$$k_v \cdot (r_{\mathrm{me}})^{\gamma_v} \cdot D_{\mathrm{target}}^{\delta_v} + E_{\mathrm{me}} \approx k_l \cdot (r_{\mathrm{llm}})^{\gamma_l} \cdot D_{\mathrm{target}}^{\delta_l} + E_{\mathrm{llm}} \tag{7}$$

Our goal is to solve for $r_{\mathrm{me}}$. We begin by isolating the term containing $r_{\mathrm{me}}$ on one side of the equation:

$$k_v \cdot (r_{\mathrm{me}})^{\gamma_v} \cdot D_{\mathrm{target}}^{\delta_v} \approx k_l \cdot (r_{\mathrm{llm}})^{\gamma_l} \cdot D_{\mathrm{target}}^{\delta_l} + (E_{\mathrm{llm}} - E_{\mathrm{me}}) \tag{8}$$

Finally, to solve for $r_{\mathrm{me}}$, we raise both sides to the power of $1/\gamma_v$, which yields the final expression:

$$r_{\mathrm{me}} \approx \left( \frac{k_l \cdot (r_{\mathrm{llm}})^{\gamma_l} \cdot D_{\mathrm{target}}^{\delta_l} + E_{\mathrm{llm}} - E_{\mathrm{me}}}{k_v \cdot D_{\mathrm{target}}^{\delta_v}} \right)^{\frac{1}{\gamma_v}} \tag{9}$$

# G  MARS IMPLEMENTATION DETAILS

## G.1  GENERAL TRAINING HYPERPARAMETERS

Unless otherwise specified, all fine-tuning experiments were conducted with the following hyperparameters. We used the Adam optimizer with its default parameter settings and a batch size of 8. The learning rate was initialized to $1 \times 10^{-5}$ and followed a cosine decay schedule with a warmup phase corresponding to 10% of the total training steps. These settings were applied consistently across all experiments, with the exception of the peak learning rate in the baseline tests for differential learning rates. Following a prior paper (Liu et al., 2023), we trained for 3 epochs on the LLaVA task and 12 epochs on the ScienceQA task, reporting the best performance.

## G.2  LoRA CONFIGURATION

For all experiments utilizing LoRA, We applied LoRA to all linear layers across the entire model, excluding the output heads. The LoRA scaling hyperparameter, `alpha`, was consistently set to twice the rank size (i.e., $\alpha = 2 \times r$). The initial calibration phase for MARS was performed using a representative set of LoRA ranks, $r \in \{8, 16, 32, 64\}$, for both the ME and LLM modules.

# H  ADDITIONAL EVALUATION RESULTS

To isolate and assess pure fine-tuning capability, we conducted a series of evaluations on "from-scratch" models, which were assembled from pre-trained unimodal components with no prior exposure to the downstream multimodal tasks. Tables 4 compares MARS against the common heuristic of using a fixed LoRA rank while manually tuning differential learning rates. Table 5 provides a more comprehensive comparison against adaptive rank baselines. Across all from-scratch models, MARS consistently outperforms baseline methods. Collectively, these results confirm that MARS's convergence-aware, multimodal search is effective at discovering superior LoRA configurations, particularly in scenarios that demand robust fine-tuning from a domain-agnostic starting point.

Table 4: Comparison with fixed-rank tuning across different learning rates. We determined the best ME learning rate (★) for each LLM learning rate ($\mathbf{lr_{llm}}$) by selecting the value that yielded either the lowest perplexity or the highest accuracy. For the LLaVA Bench, lower perplexity is better (↓), while for ScienceQA, higher accuracy is better (↑). Best in bold, second-best underlined.

| Model | Benchmark | LoRA ($\mathbf{lr_{me}}$, $\mathbf{lr_{llm}}$) | | | MARS |
|---|---|---|---|---|---|
| | | (★, 1e-5) | (★, 1e-6) | (★, 1e-7) | |
| LLaVA-OV-0.5B (Li et al., 2024)* | LLaVA Bench (↓) | 5.0011 | 5.0209 | 5.4407 | **4.9547** |
| | ScienceQA (↑) | 42.49 | 42.12 | 39.88 | **45.01** |
| LLaVA-OV-7B (Li et al., 2024)* | LLaVA Bench (↓) | 3.8701 | 4.0298 | 4.3407 | **3.5609** |
| | ScienceQA (↑) | 64.15 | 63.71 | 60.24 | **65.07** |
| Qwen2-VL-7B (Wang et al., 2024)* | LLaVA Bench (↓) | 3.7738 | 3.7949 | 3.7991 | **3.6214** |
| | ScienceQA (↑) | 65.47 | 62.87 | 59.28 | **66.47** |

Table 5: Comparison with adaptive rank search baselines. We determined the best ME rank size (★) for each LLM rank size ($\mathbf{r_{llm}}$) by selecting the value that yielded either the lowest perplexity or the highest accuracy. For the LLaVA Bench, lower perplexity is better (↓), while for ScienceQA, higher accuracy is better (↑). Best in bold, second-best underlined.

| Model | AdaLoRA | Full-rank Tuning | LoRA ($\mathbf{r_{me}}$, $\mathbf{r_{llm}}$) | | | MARS |
|---|---|---|---|---|---|---|
| | | | (★, 8) | (★, 16) | (★, 32) | |
| *LLaVA Bench (perplexity ↓)* | | | | | | |
| LLaVA-OV-0.5B (Li et al., 2024)* | 5.0807 | 5.1798 | 5.2819 | 5.0011 | 4.9912 | **4.9547** |
| LLaVA-OV-7B (Li et al., 2024)* | 4.0012 | 3.8812 | 3.6175 | 3.8701 | 3.9204 | **3.5609** |
| Qwen2-VL-7B (Wang et al., 2024)* | 4.1555 | 3.7561 | 3.7480 | 3.7738 | 4.1057 | **3.6214** |
| *ScienceQA bench (accuracy (%) ↑)* | | | | | | |
| LLaVA-OV-0.5B (Li et al., 2024)* | 41.52 | 41.46 | 38.27 | 42.49 | 41.99 | **45.01** |
| LLaVA-OV-7B (Li et al., 2024)* | 58.28 | 59.28 | 64.11 | 64.15 | 64.07 | **65.07** |
| Qwen2-VL-7B (Wang et al., 2024)* | 60.08 | 64.67 | 65.87 | 65.47 | 63.67 | **66.47** |

# I  CORRELATION BETWEEN PERPLEXITY AND FINAL TASK PERFORMANCE

In this section, we examine the relationship between convergence status and fine-tuning effectiveness. We use perplexity for this analysis because it is directly tied to the training objective (as a monotonic transformation of the loss). Lower perplexity on the test set therefore serves as a straightforward indicator of how well the model has been fine-tuned on a given dataset.

Our choice of perplexity as a task-agnostic proxy for fine-tuning quality is supported by prior work in both the LLM and MLLM literature (Li et al., 2024; Shukor et al., 2025), which shows that decreasing perplexity correlates with improved downstream task performance.

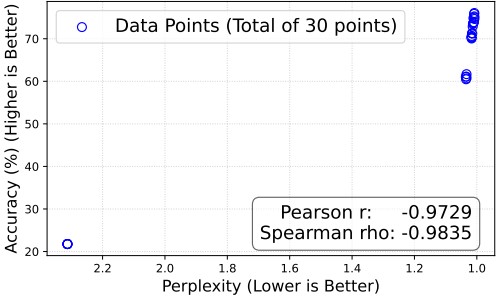

Figure 9: Validation on ScienceQA

Although recent studies (Shukor et al., 2025) have demonstrated this correlation across several MLLM benchmarks, its validation on the ScienceQA dataset has been missing. To verify this relationship on ScienceQA, Figure 9 shows a statistically significant and strongly negative correlation between perplexity and final ScienceQA accuracy (Pearson's $r = -0.97$, $p < .001$; Spearman's $\rho = -0.98$, $p < .001$). This near-perfect inverse relationship confirms that, for this dataset, perplexity is not only a theoretically motivated metric but also a reliable predictor of final task accuracy.

## J    MARS FOCUS CLARIFICATION (REBUTTAL: GENERAL RESPONSE 3)

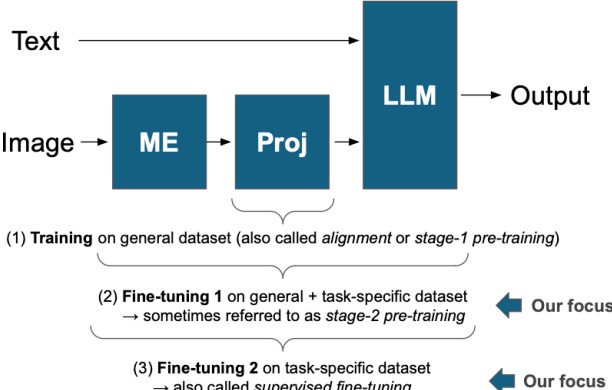

Figure 10: MARS's scope includes both fine-tuning types: generalist training and specialist adaptation.

- **[Type 1] Fine-tuning for Generalists:** Training the full model (after projector alignment) on diverse multimodal instructions to build a strong generalist foundation model (extended focus).
- **[Type 2] Fine-tuning for Specialists:** Adapting a pre-trained generalist to a specific domain dataset to obtain a specialized expert model (the primary focus of our original manuscript).

We clarify the scope of VLM fine-tuning considered in this work. In the VLM literature, fine-tuning generally falls into two categories. Our original manuscript primarily focused on Type 2 (Specialist), evaluating how effectively MARS adapts a model to a specific task with or without prior domain knowledge. However, we agree with the reviewers that Type 1 (Generalist) is equally important for establishing MARS as a robust method for developing foundational multimodal models. We appreciate the reviewers for prompting us to address this broader perspective.

## K    SIMULTANEOUS MULTI-RANK TUNING (REBUTTAL: GENERAL RESPONSE 4)

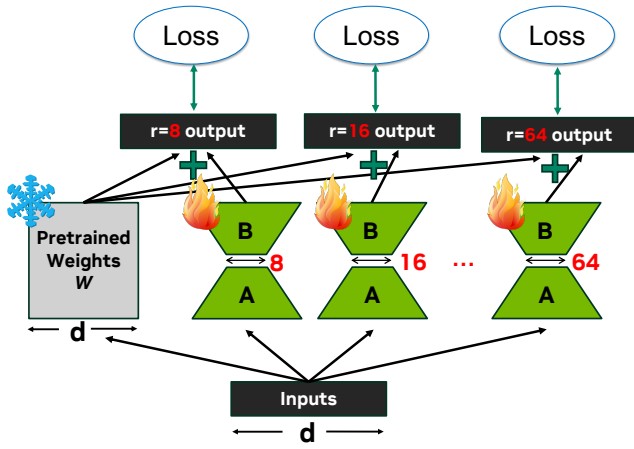

Figure 11: Simultaneous Multi-Rank Tuning Mechanism

Figure 11 illustrates our proposed optimization for the calibration phase, leveraging the frozen nature of the pre-trained backbone weights ($W$). In standard fine-tuning, the computationally expensive forward pass through the large backbone ($W$) must be recomputed for every distinct rank configuration tested. To eliminate this redundancy, we construct a computational graph where the backbone features are computed only once per batch. These shared features are then branched into multiple, lightweight parallel LoRA adapter heads (e.g., initialized with ranks $r = \{8, 16, 32, 64\}$). Since the LoRA parameters constitute a negligible fraction of the total model size ($< 1\%$), the computational overhead of running these parallel branches is minimal. This strategy allows us to gather perplexity and convergence statistics for all candidate ranks simultaneously in a single training run, in the best case, effectively reducing the calibration search cost from $\mathcal{O}(K^N)$ to $\mathcal{O}(1)$ (where $K$ is the number of rank candidates and $N$ is the number of modalities). This optimization ensures that MARS remains highly efficient even as the search space expands.

## L  THE USE OF LARGE LANGUAGE MODELS (LLMS)

For this submission, Large Language Models (LLMs) were primarily used for sentence polishing, grammar checking, and LaTeX error correction. Their use in each section is summarized below:

- **Introduction:** Improving paragraph structure.
- **MARS Methodology:** Enhancing the presentation of algorithms (e.g., colorizing, making the representation more concise).
- **Evaluation:** Debugging experiment code; refining the presentation of plots and tables; and arranging text and tables effectively in the paper.
- **Related Work:** Assisting in finding additional related works.
- **Overall:** Polishing for clarity and performing grammar checks.

