# OpenReview forum: "MARS: Harmonizing Multimodal Convergence via Adaptive Rank Search"
_ICLR.cc/2026/Conference — Submitted to ICLR 2026_

### Official Review · Reviewer_M2AU · 2025-10-26

**Soundness:** 2
**Presentation:** 3
**Contribution:** 2
**Rating:** 2
**Confidence:** 4

**Summary:**

This paper proposes MARS (Multimodal Adaptive Rank Search), a novel and efficient framework for harmonizing the imbalanced training dynamics in multimodal large language model (MLLM) fine-tuning. Recognizing that uniform LoRA ranks or heuristic learning rate tuning often lead to suboptimal performance due to mismatched convergence speeds between the modality encoder (ME) and the LLM backbone, MARS introduces a dual scaling law approach: Scaling Law-C models module-specific convergence time to prune the search space to rank pairs with aligned training dynamics, while Scaling Law-P predicts final task performance to select the optimal pair from this pruned set. By treating LoRA rank as a direct controller of modality-specific adaptation capacity and convergence speed, MARS automates the discovery of optimal, differential rank configurations. Experiments show that MARS consistently outperforms strong baselines including differential learning rates, fixed rank pairs, and AdaLoRA, improving ScienceQA accuracy by up to 12.0% and reducing LLaVA Bench perplexity by up to 13.2%, while cutting total search and fine-tuning time by over 11.5× compared to naive search.

**Strengths:**

1. The problem defined in this paper is highly valuable and carries significant practical relevance to the fine-tuning of Multimodal Large Language Models (MLLMs).
2. The two proposed scaling laws are conceptually insightful and offer meaningful inspiration for the community.
3. The paper is generally well-written, with clear and thorough explanations of the problem formulation, motivation, and methodology.

**Weaknesses:**

1. The motivation lacks strong empirical evidence to substantiate the core claims.
2. The experimental evaluation in Section 3 is relatively limited; additional results across more models, datasets, and benchmarks are needed.
3. Several technical details are insufficiently explained, reducing the reproducibility and clarity of the method.

**Questions:**

1. Regarding the motivation and empirical evidence:
a) Why do the authors believe that aligning the convergence times of the modality encoder (ME) and LLM leads to better final performance? This conclusion is not evident from Figures 2(a) and (b). A direct study correlating the gap in convergence times with final task performance would be more convincing.
b) Convergence time is highly dependent on experimental setup factors such as hardware, training framework, and distribution configurations, and may not be a stable or generalizable metric. Using aligned convergence time as a search criterion may therefore lack robustness across different deployment scenarios.
c) Figures 2(a) and (b) only report results on LLaVA-OV-0.5B. This is insufficient to support the general claim. The authors should include results from additional architectures and a broader set of rank pairs.

2. Regarding the evaluation:
a) The paper evaluates only on LLaVA-Bench and ScienceQA. Broader benchmark coverage (e.g., MME, MM-Vet, POPE, or domain-specific tasks) is necessary to demonstrate the generality of MARS.
b) In Table 1, performance consistently improves as the learning rate increases. The authors should explore higher learning rates to better characterize the performance boundary of LoRA.
c) The baselines appear somewhat outdated (e.g., AdaLoRA from 2023). Comparisons with more recent adaptive methods would strengthen the evaluation.

3. Regarding writing and technical details:
a) Beyond the ambiguity of “convergence time” the definition of convergence itself is unclear. What specific criterion or threshold is used to determine that a module has converged? This needs explicit clarification.
b) In Figure 3(a), the bar for “FT w/ MARS (final FT)” is missing for Qwen-3B, and similarly in Figure 3(b) for LLaVA-0.5B. What is the reason for this omission?
c) The ablation study states: “increasing the data sampling range up to 2^11 leads to a clear improvement,” yet the x-axis in Figure 4 only goes up to 8 (2^8 ?). This discrepancy should be resolved, and the figure should be updated to reflect the claimed range.

---

> ### Author Response · Authors · 2025-11-23
> **Part1: Q1-a/b/c & Q2-a**
>
> We greatly appreciate the issues you raised regarding motivation, experimental results, writing, and technical details. In particular, we are grateful that you framed these points as detailed, constructive questions, allowing us to clarify our perspective effectively. We have carefully addressed all your concerns below and updated the manuscript accordingly. Detailed responses are provided here, with additional context in the General Response section and the revised draft.
>
> ---
>
> **Q1-a:** Why do the authors believe that aligning the convergence times of the modality encoder (ME) and LLM leads to better final performance? This conclusion is not evident from Figures 2(a) and (b). A direct study correlating the gap in convergence times with final task performance would be more convincing.
>
> **A:** Below, We offer a two-part clarification linking convergence alignment, perplexity, and final accuracy.
>
> * **Validity of Perplexity as a Performance Proxy:** First, we clarify that we use perplexity as a robust, task-agnostic proxy for fine-tuning quality because it is directly tied to the training objective. This choice is supported by prior work in both LLM and MLLM literature [1, 2], which establishes that lower perplexity correlates with improved downstream task performance. To validate this in our fine-tuning setup, we conducted a new analysis on the ScienceQA dataset (**Appendix H**), revealing a strong negative correlation between test perplexity and final accuracy (Pearson's $r = -0.97, p < .001$). This confirms that optimizing for perplexity (our proxy) effectively maximizes final task accuracy.
>
> * **Linking Convergence Gap to Performance:** To directly address the reviewer's request, we performed a statistical correlation analysis between the Convergence Gap ($|T_{me} - T_{llm}|$) and Perplexity across multiple fine-tuning regimes ($D=512$ to $4096$). We observed statistically significant positive correlations ($r > 0.86$, $p < 0.01$), mathematically confirming that minimizing the convergence gap is a statistically dominant predictor of better performance (lower perplexity).
>
> | Dataset Size | Correlation (Pearson's r) | P-value |
> |--------------|----------------------------|---------|
> | 512          | 0.9043                     | 0.0020  |
> | 1024         | 0.9501                     | 0.0003  |
> | 2048         | 0.8683                     | 0.0052  |
> | 4096         | 0.9838                     | 0.0000  |
>
>
> * Reference**
>
> [1] Demystifying Prompts in Language Models via Perplexity Estimation (EMNLP 2023)
>
> [2] Scaling Laws for Native Multimodal Models (ICCV 2025)
>
> [3] When Scaling Meets LLM Finetuning: The Effect of Data, Model and Finetuning Method (ICLR 2024)
>
> ---
>
> **Q1-b:** Convergence time is highly dependent on experimental setup factors such as hardware, training framework, and distribution configurations, and may not be a stable or generalizable metric. Using aligned convergence time as a search criterion may therefore lack robustness across different deployment scenarios.
>
> **A:** We agree with this point. This is why we define *convergence time* strictly as the number of training steps required to satisfy an early stopping criterion, rather than wall-clock time. By positioning MARS as part of the pre-training stage, we ensure that the search and fine-tuning operate in the same environment, making the relative convergence status between the ME and LLM a robust, setup-independent metric. Please let us know if we have misunderstood your concern.
>
>
> ---
>
> **Q1-c:** Figures 2(a) and (b) only report results on LLaVA-OV-0.5B. This is insufficient to support the general claim. The authors should include results from additional architectures and a broader set of rank pairs.
>
> *A:* Please refer to **General Response 2** for additional validation on new architectures.
>
>
> ---
>
> **Q2-a:** The paper evaluates only on LLaVA-Bench and ScienceQA. Broader benchmark coverage (e.g., MME, MM-Vet, POPE, or domain-specific tasks) is necessary to demonstrate the generality of MARS.
>
> **A:** Please refer to **General Response 3** for our expanded evaluation on broader benchmarks.

---

> > ### Author Response · Authors · 2025-11-23
> > **Part2: Q2-b/c & Q3-a/b/c**
> >
> > **Q2-b:** In Table 1, performance consistently improves as the learning rate increases. The authors should explore higher learning rates to better characterize the performance boundary of LoRA.
> >
> > **A:** We thank the reviewer for this suggestion. In Table 1, we selected the candidate learning rates (`1e-5, 5e-6, 1e-6,
> > 1e-7`) based on established MLLM fine-tuning literature [1, 2] and our constraints (e.g., a small batch size of 8), where `1e-5 to 5e-5` is widely cited as the practical upper bound for stability.
> >
> > We acknowledge that we could not perform an exhaustive sweep of higher learning rates due to budget constraints. However, our primary focus is to demonstrate that MARS outperforms even the best-performing fixed LR within this standard regime. This is why the ★ values in **Table 1** differ across fine-tuning settings. Along with this, these results guided us to fix the learning rate for subsequent experiments.
> >
> > * **Reference**
> >
> > [1] Li, Bo, et al. "Llava-onevision: Easy visual task transfer." arXiv preprint arXiv:2408.03326 (2024).
> >
> > [2] Jin, Hongpeng, et al. "Rethinking learning rate tuning in the era of large language models." 2023 IEEE 5th International Conference on Cognitive Machine Intelligence (CogMI). IEEE, 2023.
> >
> > ---
> >
> > **Q2-c:** The baselines appear somewhat outdated (e.g., AdaLoRA from 2023). Comparisons with more recent adaptive methods would strengthen the evaluation.
> >
> > **A:** Please refer to **General Response 1** for comparisons with more recent adaptive methods.
> >
> >
> > ---
> > **Q3-a:** Beyond the ambiguity of “convergence time” the definition of convergence itself is unclear. What specific criterion or threshold is used to determine that a module has converged? This needs explicit clarification.
> >
> > **A:** We define *convergence* using a standard early stopping criterion based on validation performance (e.g., the lowest validation perplexity) [1]. Following PyTorch’s early stopping criterion [2], we consider the model *converged* if the validation performance does not improve for a patience period of 5 steps.
> >
> > * **Reference**
> >
> > [1] Prechelt, Lutz. "Early stopping-but when?." Neural Networks: Tricks of the trade. Berlin, Heidelberg: Springer Berlin Heidelberg, 2002. 55-69.
> >
> > [2] https://docs.pytorch.org/ignite/generated/ignite.handlers.early_stopping.EarlyStopping.html
> >
> > ---
> >
> > **Q3-b:** In Figure 3(a), the bar for “FT w/ MARS (final FT)” is missing for Qwen-3B, and similarly in Figure 3(b) for LLaVA-0.5B. What is the reason for this omission?
> >
> > **A:** Thank you for pointing this out. The values were correct; however, the previous plot was misleading because `Matplotlib`’s log-scale autoscaling placed the Y-axis lower limit near $10^1$, which compressed the bottom 10 hours of the bar and made the Qwen-3B (a) and LLaVA-0.5B (b) fine-tuning times appear negligible. We have updated **Figure 3** to use a linear scale, ensuring the full magnitude of the cost is clearly visible.
> >
> > ---
> > **Q3-c:**  The ablation study states: “increasing the data sampling range up to $2^11$ leads to a clear improvement,” yet the x-axis in Figure 4 only goes up to 8 ($2^8$ ?). This discrepancy should be resolved, and the figure should be updated to reflect the claimed range.
> >
> > **A:** We agree that the current plot notation is confusing. In **Figure 4**, the x-axis actually goes up to `8K`, and the point labeled `2 (i.e., 2K == 2^11)` marks where the curve begins to stabilize. We will update the figure to use consistent notation across the x-axis to resolve this discrepancy.

---

> ### Author Response · Authors · 2025-11-26
>
> Dear Reviewer M2AU,
>
> We hope that our responses have adequately addressed your concerns regarding **(1)** motivation and empirical evidence, **(2)** evaluation, and **(3)** writing and technical details. If you have additional questions or concerns, please let us know, and we would be happy to provide further clarification. We truly appreciate your feedback and look forward to hearing from you soon.
>
> Best regards,
>
> MARS Authors

---

### Official Review · Reviewer_iVvQ · 2025-10-31

**Soundness:** 3
**Presentation:** 2
**Contribution:** 3
**Rating:** 6
**Confidence:** 3

**Summary:**

The paper proposes MARS, a novel adaptive rank search method for parameter-efficient fine-tuning of MLLMs. MARS addresses the imbalance in training dynamics across different modalities by searching for optimal modality-specific LoRA ranks to harmonize convergence speeds. The method employs dual scaling laws—one predicting convergence speed and the other predicting final task performance—to efficiently prune and select rank pairs without exhaustive fine-tuning. Evaluation demonstrates that MARS outperforms baseline methods including differential learning rates and unimodal adaptive rank tuning, achieving superior performance and significantly reduced search time.

**Strengths:**

- The paper addresses a compelling and practical issue in MLLM fine-tuning: imbalanced training dynamics across modalities leading to suboptimal performance.

- MARS innovatively leverages dual scaling laws to guide rank search, avoiding costly exhaustive hyperparameter tuning.

- Experimental results are thorough and convincing, demonstrating consistent performance gains across multiple MLLM architectures and benchmarks including from-scratch and domain-specialized settings.

**Weaknesses:**

- The dual scaling laws are empirically derived from limited datasets and modeling scenarios; their generalizability to broader or more diverse MLLM setups remains to be established.

- The method is demonstrated on models with primarily two modality types (modality encoder and LLM), limiting insights on scalability to more complex multimodal architectures involving multiple modalities simultaneously.

**Questions:**

- Does MARS assume or require fixed LoRA configuration points for calibration? How would it perform with different or continuous rank values outside the discretized set?

- How sensitive is MARS to the initial model state, e.g., pre-trained vs. from-scratch? Are there differences in optimal rank configurations or scaling law parameters?

- Would the approach extend naturally to fine-tuning MLLMs with more than two modalities, e.g., including audio, video, or text-vision-audio tri-modal models?

---

> ### Author Response · Authors · 2025-11-23
>
> We greatly appreciate your insightful comments and suggestions. We have addressed your concerns below and incorporated the corresponding revisions into the updated manuscript. Additional details can be found in the General Response section and the revised draft.
>
> ---
>
> **W-1:** The dual scaling laws are empirically derived from limited datasets and modeling scenarios; their generalizability to broader or more diverse MLLM setups remains to be established.
>
> **A:** Please refer to **General Responses 2 and 3** for additional validation supporting the generality of MARS, including new experimental results and broader multimodal task coverage.
>
> ---
> **Q-1:** Does MARS assume or require fixed LoRA configuration points for calibration? How would it perform with different or continuous rank values outside the discretized set?
>
> **A:**
> MARS utilizes four distinct rank values (spaced to provide sufficient coverage) as anchor points during the calibration phase to fit the scaling law coefficients. Although calibrated on a discrete set, our Scaling Laws are modeled as continuous power-law functions. This formulation enables MARS to effectively interpolate and predict performance for any continuous rank value (e.g., $r=42$) that lies outside the discretized set.
>
> * **Justification:** This interpolation capability is grounded in our empirical observation of the smooth, order-preserving nature of the scaling curves (**Figure 2 and Appendices C–E**). We consistently observe smooth changes in training dynamics with no spikes or discontinuities, ensuring that the nearest discrete rank to our prediction serves as a robust approximation. By using only a few anchor points to capture this smooth landscape, MARS maintains computational efficiency without sacrificing search effectiveness.
>
> ---
> **Q-2:** How sensitive is MARS to the initial model state, e.g., pre-trained vs. from-scratch? Are there differences in optimal rank configurations or scaling law parameters?
>
> **A:**
> MARS is designed to be sensitive/adaptive to the initial model state. Our experiments reveal that the optimal rank pair {${r_{me}​,r_{llm}​}$} shifts significantly depending on model status (from-scratch vs. pre-trained), data scale, and domain gap. For from-scratch models (unaligned), the optimal configuration is highly sensitive to data scale (**Appendix D**): small datasets (2048) require high capacity for alignment ({28,32}), while larger datasets require leaner ranks ({4,8}).
>
> Conversely, pre-trained models are less sensitive to data size but sensitive to the domain gap. On the in-domain LLaVA-Bench, optimal ranks remain low and stable ({6,16} vs. {10,16}), whereas adapting to the OOD ScienceQA domain demands significantly higher capacity ({22,32}). These results demonstrate the necessity of MARS’s adaptive search.
>
> | Model Type                     | Model / Optimal Rank                               | Dataset (Characteristic)                    |
> |-------------------------------|----------------------------------------------|-----------------------------------|
> | From-Scratch Model        | LLaVA-OV-7B* / {$r_{me}$: 28, $r_{llm}$: 32}         | LLaVABench (mini: 2048)                 |
> |                               | LLaVA-OV-7B* / {$r_{me}$: 4, $r_{llm}$: 8}           | LLaVABench (full)           |
> | Pre-trained Model         | LLaVA-OV-7B / {$r_{me}$: 6, $r_{llm}$: 16}           | LLaVABench (mini: 2048)                 |
> |                               | LLaVA-OV-7B / {$r_{me}$: 10, $r_{llm}$: 16}          | LLaVABench (full)           |
> | Pre-trained Model         | LLaVA-OV-7B / {$r_{me}$: 22, $r_{llm}$: 32}          | ScienceQA (relatively long-tail)       |
>
>
> ---
> **W-2:** The method is demonstrated on models with primarily two modality types (modality encoder and LLM), limiting insights on scalability to more complex multimodal architectures involving multiple modalities simultaneously.
>
> **&**
>
> **Q-3:** Would the approach extend naturally to fine-tuning MLLMs with more than two modalities, e.g., including audio, video, or text-vision-audio tri-modal models?
>
> **A:**  Please refer to **General Response 4** for a detailed explanation of the scalability of MARS to $N>2$ modalities.

---

> > ### Author Response · Authors · 2025-11-26
> >
> > Dear Reviewer iVvQ,
> >
> > We hope that our responses have adequately addressed your concerns regarding **(1)** the empirical nature and generalizability of the scaling laws of our scaling laws, **(2)** the scalability of MARS to complex architectures with more than two modalities, and **(3)** the sensitivity of the method to continuous rank values and varying initial model states. If you have additional questions or concerns, please let us know, and we would be happy to provide further clarification. We truly appreciate your feedback and look forward to hearing from you soon.
> >
> > Best regards,
> >
> > MARS Authors

---

> > > ### Comment · Reviewer_iVvQ · 2025-11-27
> > >
> > > Thanks for the response. I do hope authors can take weaknesses into the revision carefully. I will arise my rating accordingly.

---

### Official Review · Reviewer_H596 · 2025-11-01

**Soundness:** 3
**Presentation:** 3
**Contribution:** 3
**Rating:** 6
**Confidence:** 2

**Summary:**

This paper introduces MARS, a method to address imbalanced convergence speeds in fine-tuning multimodal large language models (MLLMs). By leveraging dual scaling laws, MARS predicts module-specific convergence and prunes the LoRA rank search space, efficiently identifying optimal rank pairs prior to full fine-tuning. Experiments demonstrate that MARS achieves superior performance over baseline heuristic strategies, such as differential learning rates or naive rank selection, across multiple tasks, including ScienceQA and LLaVA Bench.

**Strengths:**

The proposed method is experimentally validated, showing consistent improvements in task performance across different LLMs. It clearly outperforms naive and heuristic approaches. Mostly easy to follow.

**Weaknesses:**

The scaling laws are empirically derived from a limited set of tasks (ScienceQA, LLaVA Bench); it remains unclear how well they generalize to other multimodal tasks or broader modality combinations.

**Questions:**

- Is there an impact of MARS on model robustness or generalization to out-of-distribution multimodal inputs?
- Can the method scale effectively for MLLMs with more than two modalities, or in scenarios with limited pretraining alignment across modalities?

---

> ### Author Response · Authors · 2025-11-23
>
> Thank you for your thoughtful comments and suggestions, which helped improve our paper. We have responded to your concerns below and updated the manuscript accordingly. Further details are included in the General Response section and the revised draft.
>
> ---
> **W-1:**  The scaling laws are empirically derived from a limited set of tasks (ScienceQA, LLaVA Bench); it remains unclear how well they generalize to other multimodal tasks or broader modality combinations.
>
> **A:** Please refer to **General Responses 2 and 3** for additional validation supporting the generality of MARS, including further results and broader multimodal task coverage.
>
> ---
> **Q-1:**  Is there an impact of MARS on model robustness or generalization to out-of-distribution multimodal inputs?
>
> **A:** As shown in **General Response 3**, MARS exhibits superior robustness and generalization to out-of-distribution inputs. Specifically, when trained on the LLaVA-1.5 dataset, MARS consistently outperforms baselines on benchmarks such as GQA, TextCaps, and AI2D, which involve diagram and OCR data not heavily represented in the training distribution.
>
> ---
> **Q-2:** Can the method scale effectively for MLLMs with more than two modalities, or in scenarios with limited pretraining alignment across modalities?
>
> **A:** Please refer to **General Response 4** for a detailed discussion on scalability to $N>2$ modalities.
>
> Regarding scenarios with limited pre-training alignment, we interpret this as cases where the modalities are not yet well-synchronized (e.g., initialization from separate unimodal encoders). Our experiments with 'from-scratch' models demonstrate that MARS is highly effective in this regime (**see Appendix H**). It automatically detects the large convergence gap caused by misalignment and assigns appropriate rank capacities to bridge it, whereas heuristic methods fail to capture this dynamic need.

---

> > ### Author Response · Authors · 2025-11-26
> >
> > Dear Reviewer H596,
> >
> > We hope that our responses have adequately addressed your concerns regarding **(1)** the empirical nature and generalizability of the scaling laws of our scaling laws, **(2)** the impact of MARS on model robustness and out-of-distribution inputs, and **(3)** the scalability of the method to MLLMs with more than two modalities. If you have additional questions or concerns, please let us know, and we would be happy to provide further clarification. We truly appreciate your feedback and look forward to hearing from you soon.
> >
> > Best regards,
> >
> > MARS Authors

---

### Official Review · Reviewer_jcpL · 2025-11-01

**Soundness:** 3
**Presentation:** 3
**Contribution:** 3
**Rating:** 6
**Confidence:** 3

**Summary:**

The paper introduces MARS a two-stage procedure that automates modality-specific LoRA-rank selection for MLLM fine-tuning by combining two compact scaling laws. A closed-form solution finds the balanced ME-rank for a given LLM-rank. The laws are calibrated with a few short training runs. On models like LLaVA and Qwen2.5-VL, MARS outperforms baselines (tuned LRs, fixed ranks, AdaLoRA) on LLaVA-Bench and ScienceQA. It also cuts the total search and fine-tuning time by 11.5-15x compared to a 4x4 grid search.

**Strengths:**

* Replaces a brute-force grid search with an efficient "prune-then-predict" strategy and a closed-form solution (Eq. 3) for balancing ranks.

* The log-log plots (Fig. 2b, 6) show near-parallel lines, supporting the formula's assumption that rank and data size effects are separable.

* Achieves better perplexity and accuracy than baselines on LLaVA-OV-7B and other models.

* Cuts search and training time by over 11.5x versus a simple 4x4 grid search.

* Still shows gains even when tested on "from-scratch" models (Table 3).

**Weaknesses:**

* Results focus on "search time" savings, but higher ranks (which MARS might pick) cost more per-step in FLOPs/memory. The total end-to-end wall-clock/energy cost isn't compared.

* Why only LoRA on q/k/v? The projector is tuned but not part of the rank search. Ranks are per-module, not per-layer. Rounding the continuous rank from Eq. 3 to the nearest discrete one seems crude; a local sweep wasn't tested.

* The scaling exponents are fit on "from-scratch" models but then used for pre-trained MLLMs. How stable are these exponents across different models, seeds, or domains?

**Questions:**

* Is "convergence" based on the validation set or test set? How do you ensure the calibration phase doesn't peek at the test set?

* Could this same two-law framework be used to optimize the projector rank or to find layer-specific ranks?

* Do the scaling exponents generalize to other tasks (like OCR or chart VQA), or must they be re-fit for every new domain?

---

> ### Author Response · Authors · 2025-11-23
>
> We sincerely appreciate your constructive feedback. We have addressed all your comments below and incorporated corresponding revisions into the updated manuscript. Further details are provided in the General Response section and the revised draft.
>
> ---
>
> **W-1:** Results focus on "search time" savings, but higher ranks (which MARS might pick) cost more per-step in FLOPs/memory. The total end-to-end wall-clock/energy cost isn't compared.
>
> **A:** The total end-to-end wall-clock time (including both search and fine-tuning) is explicitly compared in **Figure 3**. As shown, MARS achieves a reduction in total time despite the search overhead. Please let us know if we have misunderstood your specific concern regarding cost accounting.
>
> ---
>
> **W-2:** Why only LoRA on q/k/v? The projector is tuned but not part of the rank search. Ranks are per-module, not per-layer. Rounding the continuous rank from Eq. 3 to the nearest discrete one seems crude; a local sweep wasn't tested.
>
> **A:**
>
> * **(1) LoRA Targets:** We sincerely apologize for the error in **Appendix G.2**, which incorrectly stated that LoRA was applied to limited modules. In our actual implementation (consistent with QLoRA [1]), we applied LoRA to all linear layers in the transformer blocks, excluding the LM head. This has been corrected in the revision.
>
>  * **(2) Projector:** We treat the projector as part of the modality encoder because they process the same input stream for the same semantic purpose.
>
> * **(3) Rounding & Local Sweep:** Our decision to round is justified by the order-preserving nature of the scaling curves consistently observed in our experiments (**Figure 2 and Appendices C–E**). Based on these observations, which show smooth changes with no spikes in the relationship between rank and training dynamics, we believe that simple rounding is an optimal choice while maintaining the efficiency and effectiveness of MARS.
>
>
> **Reference**
>
> [1] Dettmers, Tim, et al. "Qlora: Efficient finetuning of quantized llms." Advances in neural information processing systems 36 (2023): 10088-10115.
>
> ---
>
> **W-3:** The scaling exponents are fit on "from-scratch" models but then used for pre-trained MLLMs. How stable are these exponents across different models, seeds, or domains?
>
> **A:** First, we clarify that we do not apply ‘from-scratch’ exponents to pre-trained models. For **Tables 1 and 2**, the exponents come from calibration with the target pre-trained model, fitting the scaling laws to the current fine-tuning setup. This is why MARS is designed as part of the pre-training stage: to ensure that the scaling laws are calibrated in the same environment (model state, seed, data) as the subsequent fine-tuning. Regarding stability across domains, please refer to **General Response 3**, which demonstrates MARS’s robustness.
>
> ---
>
> **Q-1:** Is "convergence" based on the validation set or test set?
>
> **A:** We define convergence using a standard early stopping criterion based on validation performance (e.g., the lowest validation perplexity) [1]. Following PyTorch’s early stopping criterion [2], we consider the model converged if the validation metric does not improve for a patience period of 5 steps.
>
> * **Reference**
>
> [1] Prechelt, Lutz. "Early stopping-but when?." Neural Networks: Tricks of the trade. Berlin, Heidelberg: Springer Berlin Heidelberg, 2002. 55-69.
>
> [2] https://docs.pytorch.org/ignite/generated/ignite.handlers.early_stopping.EarlyStopping.html
>
> ---
>
> **Q-2:** Could this same two-law framework be used to optimize the projector rank or to find layer-specific ranks?
>
> **A:** Regarding projector rank, it is handled as part of the modality encoder block (see **W-2**).
>
> Regarding layer-specific ranks, our current work focuses on the macroscopic module-level imbalance between heterogeneous components. While extending this framework to simultaneously resolve module imbalance and assign fine-grained layer ranks is a compelling future direction, we position MARS as the foundational step enabling such advanced exploration.
>
> ---
> **Q-3:** Do the scaling exponents generalize to other tasks (like OCR or chart VQA), or must they be re-fit for every new domain?
>
> **A:** Please refer to **General Response 3** for our expanded evaluation on *Generalist* capabilities. Our results show that MARS is highly robust to domain shifts: trained on LLaVA-1.5 data, it still outperforms baselines on OOD tasks like GQA, TextCaps (OCR), and AI2D (Diagrams), confirming that the learned rank balance generalizes effectively.

---

> ### Author Response · Authors · 2025-11-26
>
> Dear Reviewer jcpL,
>
> We hope that our responses have adequately addressed your concerns regarding **(1)** the analysis of total end-to-end wall-clock time, **(2)** the rationale behind our module selection (projector/layer-specific) and rounding strategies, and **(3)** the generalization and stability of the scaling exponents across different domains. If you have additional questions or concerns, please let us know, and we would be happy to provide further clarification. We truly appreciate your feedback and look forward to hearing from you soon.
>
> Best regards,
>
> MARS Authors

---

### Official Review · Reviewer_nHTE · 2025-11-01

**Soundness:** 3
**Presentation:** 3
**Contribution:** 3
**Rating:** 4
**Confidence:** 3

**Summary:**

This paper introduces MARS (Multimodal Adaptive Rank Search), an approach for automatically discovering optimal differential LoRA rank pairs for fine-tuning MLLMs. The authors posit that imbalanced training dynamics between modality encoder and LLM modules, caused by fixed rank assignments, result in suboptimal convergence and performance. The proposed MARS utilizes dual scaling laws—one for module-specific convergence speed (Scaling Law-C) and another for final task performance (Scaling Law-P)—to efficiently prune and search the candidate space of LoRA ranks. The method is evaluated under various MLLM configurations and benchmark datasets, showing improved accuracy and efficiency.

**Strengths:**

1.	The idea of using dual scaling laws to guide rank selection for multimodal fine-tuning is innovative and gains a clear reduction in computational burden for hyperparameter search.
2.	The proposed dual scaling laws are empirically calibrated and validated with extensive experiments.
3.	Experimental results over multiple baselines across two standard multimodal benchmarks demonstrate consistent improvements.

**Weaknesses:**

1.	The related work section does not address several directly pertinent recent efforts on adaptive rank selection in LoRA and multimodal search. It's better to take more relevant baselines and benchmarks for comparison.
2.	The scaling laws are empirically fitted without strong theoretical justification, which may limit interpretability and generalizability.

**Questions:**

1.	How does MARS perform in low-data regimes or on tasks with very large domain shifts not seen during calibration?
2.	Can the authors provide additional evidence or theoretical insights supporting the generalizability of the proposed scaling laws to models with more than two modalities?

---

> ### Author Response · Authors · 2025-11-23
>
> We genuinely appreciate the constructive feedback, which has significantly strengthened our paper. We have addressed all questions and concerns below, and updated the manuscript accordingly. Further details are provided in the General Response section and the revised draft.
>
> ---
> **W-1:** The related work section does not address several directly pertinent recent efforts on adaptive rank selection in LoRA and multimodal search. It's better to take more relevant baselines and benchmarks for comparison.
>
> **A:** Please refer to **General Response 1** for a discussion on additional baselines, and **General Response 3** for our expanded benchmark results.
>
> ---
> **W-2:** The scaling laws are empirically fitted without strong theoretical justification, which may limit interpretability and generalizability.
>
> **A:** Please refer to **General Response 2** for additional validation supporting the generality of MARS.
>
> ---
> **Q-1:** How does MARS perform in low-data regimes or on tasks with very large domain shifts not seen during calibration?
>
> **A:** In low-data regimes (dataset size <$10^2$), our analysis (**Appendices C and D**) indicates that higher LoRA ranks are necessary to compensate for limited samples. MARS is capable of adapting to this requirement by identifying the need for higher capacity.
>
> Regarding large domain shifts, please refer to **General Response 3**. MARS consistently outperforms baselines on benchmarks such as GQA, TextCaps, and AI2D, which include diagram or OCR data that are not heavily represented in the training distribution.
>
> ---
>
> **Q-2:** Can the authors provide additional evidence or theoretical insights supporting the generalizability of the proposed scaling laws to models with more than two modalities?
>
> **A:** Please refer to **General Response 4** for a detailed explanation of the scalability of MARS to $N>2$ modalities.

---

> > ### Author Response · Authors · 2025-11-26
> >
> > Dear Reviewer nHTE,
> >
> > We hope that our responses have adequately addressed your concerns regarding **(1)** additional baselines and benchmark comparisons, **(2)** the empirical nature and generalizability of the scaling laws, and **(3)** performance in low-data or large domain-shift settings as well as **(4)** the extension to more than two modalities. If you have additional questions or concerns, please let us know, and we would be happy to provide further clarification. We truly appreciate your feedback and look forward to hearing from you soon.
> >
> > Best regards,
> >
> > MARS Authors

---

### Author Response · Authors · 2025-11-23
**General Responses 1 & 2: New Baseline & Empirical Nature of Scaling Laws**

Dear Reviewers,

We sincerely thank all reviewers (**M2AU, iVvQ, H596, jcpL, nHTE**) for highlighting many insightful points. In this General Response, we provide detailed answers to the common questions raised across the reviews. Please take a look, and feel free to reach out if anything is unclear or if you have additional questions.

---
## General Response 1: Compare with more recent baseline algorithms.

We identified two recent adaptive LoRA methods, GeoLoRA (ICLR 2025) [1] and SalientLoRA (NeurIPS 2024) [2]. Due to the constraints of the rebuttal period and code availability, we prioritized GeoLoRA for our additional baseline comparison. GeoLoRA employs low-rank geometry and matrix differential equations to adaptively determine rank structures in LLMs. Although GeoLoRA applies its algorithm to multimodal models such as ViT, their evaluation does not consider the multimodal models or fine-tuning setups that naturally create imbalanced training dynamics (caused by the two-fold disparity we introduced). Below are the results we obtained.

| Model                            |  Dataset  | AdaLoRA | GeoLoRA | LoRA (Best) | MARS   |
|-------------------------------------|---------|---------|---------|------------------|--------|
| Qwen2.5-VL-3B    |  LLaVA Bench (lower is better) | 3.6679  |     3.6676    |      3.5912     | **3.5825** |
| Qwen2.5-VL-7B        |  LLaVA Bench (lower is better) | 3.6394  |    3.6679      |      3.5011       |  **3.3616**  |
| Qwen2.5-VL-3B    |  ScienceQA  (higher is better)  | 70.06   |    70.46     |     78.24       | **79.24** |
| Qwen2.5-VL-7B   |  ScienceQA   (higher  is better) | 73.85   |     72.85    |    **79.84**     | 79.64 |

* **Reference**

[1] Schotthöfer, Steffen, et al. "GeoLoRA: Geometric integration for parameter efficient fine-tuning." arXiv preprint arXiv:2410.18720 (2024).

[2] Ke, Wenjun, et al. "Unveiling lora intrinsic ranks via salience analysis." Advances in Neural Information Processing Systems 37 (2024): 131575-131595.

---
## General Response 2: On the Empirical Nature of Scaling Laws and the Need for Additional Results

We fully agree that, since Scaling Law-P and Scaling Law-C are inherently empirical, additional experiments across a broader set of architectures and datasets are necessary to strengthen the generality of our claims. In response, we have conducted further validation and added new plots covering additional architectures, datasets, and rank pairs in **Appendices D & E**. We will also ensure that more of these plots are included in the next revision of the paper.

---

### Author Response · Authors · 2025-11-23
**General Responses 3 & 4: Broader Generality & Scalability to N-Modalities**

## General Response 3: Generality of MARS with Broader Benchmark Coverage

Before presenting our new benchmark results, we would like to clarify the scope of VLM fine-tuning discussed in this work. In the VLM area, fine-tuning generally falls into two categories (**see Figure 9 in Appendix J**).

* **[Type 1]** Fine-tuning for Generalists: Training the entire model (after projector alignment) on a diverse set of multimodal instructions to build a strong foundational generalist.

* **[Type 2]** Fine-tuning for Specialists: Adapting a pre-trained generalist to a specific domain dataset to create a specialized model.

Our original manuscript primarily focused on Type 2 (Specialist), evaluating how well MARS adapts models to specific tasks with or without prior domain knowledge. However, we agree with the reviewers that Type 1 (Generalist) is also critical for establishing MARS as a robust method for building foundational multimodal models.

**[Updated on Dec 3rd with further fine-tuning results]** Our new experiments on LLaVA-OV-7B* (without prior multimodal task knowledge) using the LLaVA-1.5-665K dataset [1] demonstrate that MARS is effective for the Type-1 case as well. With 8-epoch fine-tuning, MARS outperforms baselines on comprehensive benchmarks (MME, POPE, MMStar) and OOD tasks (GQA, TextCaps, AI2D).

| Method | MME (sum) | POPE | MMStar | TextCaps (ROUGE-L) | AI2D | GQA |
| :--- | :---: | :---: | :---: | :---: | :---: | :---: |
| AdaLoRA | 1524.0 | 79.7 | 54.7 | 38.0 | 45.5 | 52.7 |
| LoRA | 1662.4 | **86.8** | 55.4 | 42.6 | 48.0 | 55.1 |
| **MARS** | **1695.6** | 86.3 | **58.3** | **42.9** | **49.9** | **55.7** |

* **Reference**

[1] Improved Baselines with Visual Instruction Tuning (2024)

---
## General Response 4: Scalability

We confirm that MARS scales linearly ($O(N)$) with the number of modalities, whereas naive search grows exponentially. Furthermore, we propose an implementation optimization that makes the calibration phase highly parallel.

**1. Structural Scalability (The $O(N)$ Advantage):**

As illustrated in Algorithm 1 in our manuscript, naive search relies on nested loops to explore all rank combinations. For an MLLM with $N$ modalities (e.g., Audio, Video, Depth, Text), this creates a combinatorially explosive search space of size $|R_{candidates}|^N$.
However, MARS decouples the search, while transforming the problem from an $N$-dimensional combinatorial grid to $(N−1)$ independent 1D equations.

* Anchor (Outer Loop): We anchor the LLM ($r_{llm}$) as the central reference.
* Alignment (Inner Process): For a fixed $r_{llm}$, we do not need to combinatorially search other ranks. Instead, for each additional modality $i$, we simply solve the Scaling Law-C equation to find the rank pairs that satisfy the balance condition: $T_i(r_i) \approx T_{llm}(r_{llm})$.

**2. Calibration Efficiency:**

We introduce a Simultaneous Multi-Rank Adapter Tuning strategy (**Appendix K**), leveraging the frozen backbone to test multiple ranks in a single pass. This reduces calibration overhead to near-constant time ($O(1)$).

In conclusion, with linear search complexity and constant-time calibration, MARS demonstrates strong scalability potential.

---

### Meta-Review · Area_Chair_fy1h · 2026-01-08

**Summary:**

1. Motivation and Empirical Evidence:
The reviewer questioned the motivation behind aligning convergence times between modality encoders (ME) and LLMs, noting that while the paper claims this leads to better performance, the evidence was not strongly supported.

2. Scalability and Generalization:
The scalability of MARS to models with more than two modalities and its robustness across diverse multimodal tasks were questioned.

3. Technical Details and Clarity:
A few technical details were highlighted as insufficiently explained, reducing the clarity of the method and potentially hindering reproducibility.

**Reviewer Concerns:**

Outstanding Concerns:

1. Scalability to More Complex Multimodal Architectures:
While the rebuttal demonstrated the method's effectiveness with a range of benchmarks and additional modalities, there is still some lingering concern about MARS’s performance on models with more complex multimodal architectures.

2. Generalization of the Scaling Laws:
The reviewer raised concerns about the generalizability of the scaling laws, especially across models with significantly different characteristics. There is still some uncertainty about how well these scaling laws will generalize across even more diverse multimodal models.

**Reviewer Scores:**

None

---

### Decision · Program_Chairs · 2026-01-26

Reject